# Voting-based Approaches for Differentially Private Federated Learning

## Abstract

While federated learning (FL) enables distributed agents to collaboratively train a centralized model without sharing data with each other, it fails to protect users against inference attacks that mine private information from the centralized model. Thus, facilitating federated learning methods with differential privacy (DPFL) becomes attractive. Existing algorithms based on privately aggregating clipped gradients require many rounds of communication, which may not converge, and cannot scale up to large-capacity models due to explicit dimension-dependence in its added noise. In this paper, we adopt the knowledge transfer model of private learning pioneered by Papernot et al. (2017; 2018) and extend their algorithm *PATE*, as well as the recent alternative *PrivateKNN* (Zhu et al., 2020) to the federated learning setting. The key difference is that our method privately aggregates the *labels* from the agents in a *voting scheme*, instead of aggregating the *gradients*, hence avoiding the dimension dependence and achieving significant savings in communication cost. Theoretically, we show that when the margins of the voting scores are large, the agents enjoy exponentially higher accuracy and stronger (data-dependent) differential privacy guarantees on both agent-level and instance-level. Extensive experiments show that our approach significantly improves the privacy-utility trade-off over the current state-of-the-art in DPFL.

## 1 Introduction

With increasing ethical and legal concerns on leveraging private data, federated learning (McMahan et al., 2017) (FL) has emerged as a paradigm that allows agents to collaboratively train a centralized model without sharing local data. In this work, we consider two typical settings of federated learning: (1) Local agents are in large number, i.e., learning user behavior over many mobile devices (Hard et al., 2018). (2) Local agents are in small number with sufficient instances, i.e., learning a health related model across multiple hospitals without sharing patients' data (Huang et al., 2019).

When implemented using secure multi-party computation (SMC) (Bonawitz et al., 2017), federated learning eliminates the need for any agent to share its local data. However, it does not protect the agents or their users from inference attacks that combine the learned model with side information. Extensive studies have established that these attacks could lead to blatant reconstruction of the proprietary datasets (Dinur & Nissim, 2003) and identification of individuals (a legal liability for the participating agents) (Shokri et al., 2017). Motivated by this challenge, there had been a number of recent efforts (Truex et al., 2019b; Geyer et al., 2017; McMahan et al., 2018) in developing federated learning methods with differential privacy (DP), which is a well-established definition of privacy that provably prevents such attacks.

Among the efforts, DP-FedAvg (Geyer et al., 2017; McMahan et al., 2018) extends the NoisySGD method (Song et al., 2013; Abadi et al., 2016) to the federated learning setting by adding Gaussian noise to the clipped accumulated gradient. The recent state-of-the-art DP-FedSGD (Truex et al., 2019b) is under the same framework but with per-sample gradient clipping. A notable limitation for these gradient-based methods is that they require clipping the magnitude of gradients to $\tau$ and adding noise proportional to $\tau$ to *every coordinate* of the shared global model with $d$ parameters. The clipping and perturbation steps introduce either large bias (when $\tau$ is small) or large variance (when $\tau$ is large), which interferes the SGD convergence and makes it hard to scale up to large-capacity models. In Sec. 3, we concretely demonstrate these limitations with examples and theory.

Particularly, we show that the FedAvg may fail to decrease the loss function together with gradient clipping, and DP-FedAvg requires many outer-loop iterations (i.e., many rounds of communication to synchronize model parameters) to converge under differential privacy.

To avoid the gradient clipping, we propose to conduct the aggregation over the label space, as shown to be an effective approach in standard (non-federated) learning settings, i.e., voting-based model-agnostic approaches (Papernot et al., 2017; 2018; Zhu et al., 2020). To achieve it, we relax the traditional federated learning setting to allow unlabeled public data at the server side. We also consider a more complete scenario for federated learning, where there are a large number of local agents or a limited number of local agents. The agent-level privacy as introduced in DP-FedAvg, works seamlessly with our setting having many agents. However, when there are few agents, hiding each data belonging to one specific agent becomes burdensome or unnecessary. To this end, we provide a more complete privacy notion, i.e., agent-level and instance-level. Under each of the setting, we theoretically and empirically show that the proposed label aggregation method effectively removes the sensitivity issue caused by gradient clipping or noise addition, and achieves favorable privacy-utility trade-off compared to other DPFL algorithms.

Our contributions are summarized as the following:

1. We propose two voting-based DPFL algorithms via label aggregation (*PATE-FL* and *Private-KNN-FL*) and demonstrate their clear advantages over gradient aggregation based DPFL methods (e.g., *DP-FedAvg*) in terms of communication cost and scalability to high-capacity models.
2. We provide provable differential privacy guarantees under two levels of granularity: agent-level DP and instance-level DP. Each is natural in a particular regime of FL depending on the number of agents and the size of their data.
3. Extensive evaluation demonstrates that our method improves the privacy-utility trade-off over randomized gradient-based approaches in both agent-level and instance-level cases.

**A remark of our novelty.** Though *PATE-FL* and *Private-kNN-FL* are algorithmically similar to the original *PATE* (Papernot et al., 2018) and *Private-KNN* (Zhu et al., 2020), they are not the same and we are adapting them to a new problem — *federated learning*. The adaptation itself is nontrivial and requires substantial technical innovations. We highlight three challenges below.

- Several key DP techniques that contributed to the success of PATE and Private-KNN in the standard setting are no longer applicable (e.g., Privacy amplification by Sampling and Noisy Screening). This is partly due to that in standard private learning, the attacker only sees the final models; but in FL, the attacker can eavesdrop in all network traffic.

- Moreover, PATE and Private-kNN only provide instance-level DP. We show PATE-FL and Private-kNN-FL also satisfy the stronger agent-level DP. PATE-FL's agent-level DP parameter is, surprisingly, a factor of 2 better than its instance-level DP parameter. And Private-kNN-FL in addition enjoys a factor of $k$ amplification for the instance-level DP.

- A key challenge of FL is the data heterogeneity of individual agents, while PATE randomly splits the dataset so each teacher is identically distributed. The heterogeneity does not affect our privacy analysis but does make it unclear whether PATE would work. We are the first to report strong empirical evidence that the PATE-style DP algorithms remain highly effective in the non-iid case.

## 2 PRELIMINARY

In this section, we start with introducing the typical notations of federated learning and differential privacy. Then, two randomized gradient-based baselines, DP-FedAvg and DP-FedSGD, are introduced as the DPFL background.

### 2.1 FEDERATED LEARNING

Federated learning (McMahan et al., 2017; Bonawitz et al., 2017; Mohassel & Zhang, 2017; Smith et al., 2017) is a distributed machine learning framework that allows clients to collaboratively train a global model without sharing local data. We consider $N$ agents, each agent $i$ has $n_i$ data kept locally

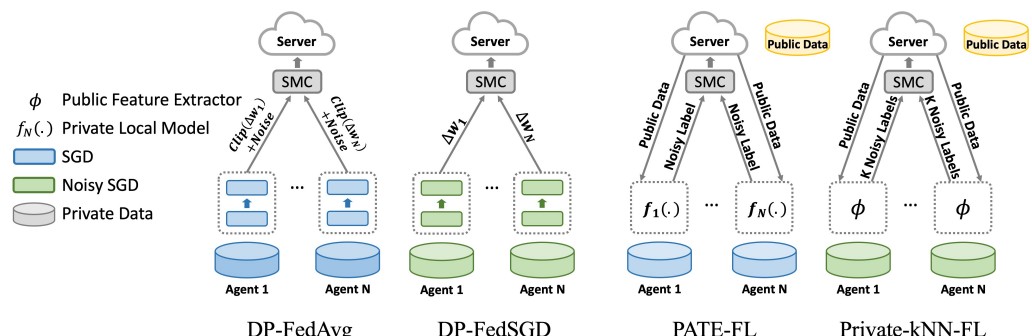

Figure 1: DP-FedAvg and *PATE-FL* are used for agent-level DP. DP-FedSGD and *Private-kNN-FL* are used for instance-level DP.

and privately from a party-specific domain distribution $\mathcal{D}_i$. $C$ is the number of classes. The objective is to output a global model that performs well on the target (server) distribution. Most prior works consider the target distribution as a uniform distribution over the union of all local data, which is restrictive in practice. Here we consider an agnostic federated learning scenario (Mohri et al., 2019; Peng et al., 2019c), where the server distribution $\mathcal{D}_G$ can be different from all agent distributions. In light of this, we assume each agent has access to part of unlabeled server data drawn from the target distribution $\mathcal{D}_G$.

FedAvg (McMahan et al., 2017) is a vanilla federated learning algorithm that we consider as a non-DP baseline. In this algorithm, a fraction of agents is sampled at each communication round with a probability $q$. Each selected agent downloads the shared global model and improves it by learning from local data using $E$ iterations of stochastic gradient descent (SGD). We denote this local update process as an inner loop. Only the gradient is sent to the server, where it is averaged with other selected agents' gradient to improve the global model. The global model is learned after $T$ communication rounds, where each communication round is denoted as one outer loop.

## 2.2 DIFFERENTIAL PRIVACY FOR FEDERATED LEARNING

Differential privacy (Dwork et al., 2006) is a quantifiable and composable definition of privacy that provides provable guarantees against identification of individuals in a private dataset.

**Definition 1.** *A randomized mechanism* $\mathcal{M} : \mathcal{D} \rightarrow \mathcal{R}$ *with a domain* $\mathcal{D}$ *and range* $\mathcal{R}$ *satisfies* $(\epsilon, \delta)$-*differential privacy, if for any two* adjacent *datasets* $D, D' \in \mathcal{D}$ *and for any subset of outputs* $S \subseteq \mathcal{R}$, *it holds that* $\Pr[\mathcal{M}(D) \in S] \leq e^\epsilon \Pr[\mathcal{M}(D') \in S] + \delta$.

The definition applies to a variety of different granularity, depending on how the *adjacent* datasets are defined, i.e., if we are to protect whether one agent participates into training, the neighboring datasets are defined by adding or removing the entire local data within that agent. It is known as agent-level (user-level) differential privacy, which has been investigated in DP-FedAvg (Geyer et al., 2017; McMahan et al., 2018). Compared to FedAvg, DP-FedAvg (Figure 1) enforces clipping of per-agent model gradient to a threshold $S$ and adds noise to the scaled gradient before it is averaged at the server. Note that this DP notion is favored when data samples within one agent reveal the same sensitive information, e.g., cell phone agents send the same message.

However, when there are only a few agents, hiding the entire dataset from one agent becomes difficult and inappropriate. We then consider the instance-level DP, where the adjacent dataset is defined by differing one single training example. This definition is consistent with the standard non-federated learning differential privacy (Abadi et al., 2016; Bassily et al., 2014; Chaudhuri et al., 2011). Model training with instance-level DP restricts the adversary's power in detecting a specific training instance's presence or absence. DP-FedSGD (Truex et al., 2019a; Peterson et al., 2019), one such state-of-the-art for the instance-level DP, performs NoisySGD (Abadi et al., 2016) for a fixed number of iterations at each agent. The gradient updates are averaged on each communication round at the server, as shown in Figure 1.

SMC is a cryptographic technique that securely aggregates local updates before the server receives it. While SMC does not have a differential privacy guarantee, it can be combined with DP to *amplify* the privacy guarantee (Bhowmick et al., 2018; Agarwal et al., 2018; Truex et al., 2019b) against attackers

that eavesdrop what sent out by each agent. In our experiment, we assume that the aggregation is conducted by SMC for all privacy-preserving algorithms that we consider.

# 3 CHALLENGES FOR GRADIENT-BASED FEDERATED LEARNING

In this section, we highlight the main challenges of the conventional DPFL frameworks in terms of accuracy, convergence and communication cost. For other challenges, we refer the readers to a survey (Kairouz et al., 2019). The details of DP-FedAvg are summarized in appendix algorithm section.

## 3.1 CHALLENGE 1: BIASED GRADIENT ESTIMATION

Recent works (Li et al., 2018) have shown that the FedAvg may not converge well under heterogeneity (e.g., non-identical distributions). Here, we provide a simple example to show that the clipping step of DP-FedAvg may raise additional challenge.

**Example 2** (clipping). *Let $N = 2$, each agent $i$'s local update is $\triangle_i$ ($E$ iterations of SGD). We enforce clipping of per-agent update $\triangle_i$ by performing $\triangle_i / \max(1, \frac{||\triangle_i||_2}{\tau})$, where $\tau$ is the clipping threshold. Consider the special case when $||\triangle_1||_2 = \tau + \alpha$ and $||\triangle_2||_2 \leq \tau$. Then the global update will be $\frac{1}{2}(\frac{\tau \triangle_1}{||\triangle_1||_2} + \triangle_2)$, which is biased.*

The unbiased global update shall be $\frac{1}{2}(\triangle_1 + \triangle_2)$. Such a simple example can be embedded in more realistic problems, causing substantial bias that leads to non-convergence.

## 3.2 CHALLENGE 2: SLOW CONVERGENCE

Recent works (Li et al., 2019; Wang et al., 2019) have investigated the convergence rate in FL methods. Here, we draw connections to DP-FedAvg's convergence rate and demonstrate that using many outer-loop iterations ($T$) could have a similar convergence issue under differential privacy.

When $E = 1$ in the local update (inner loop), the FedAvg algorithm is equivalent to SGD with distributed data, which requires many rounds of communication. The appeal of FedAvg is to set $E$ to be larger so that each agent performs $E$ iterations to update its own parameters before synchronizing the parameters to the global model, hence reducing the number of rounds in communication. However, setting $E > 1$ may not improve convergence at all.

Now, we take a closer look at the effect of increasing $E$ in the case of piecewise linear functions. Let $\eta$ be the learning rate for individual agents. In appendix convergence section, we establish that the effect of increasing $E$ is essentially increasing the learning rate for a large family of optimization problems with piecewise linear objective functions. It is known that for the family of $G-$Lipschitz functions supported on a $B$-bounded domain, any Krylov-space method [1] has a rate of convergence that is lower bounded by $\Omega(BG/\sqrt{T})$ (Nesterov, 2003, Section 3.2.1). This indicates that the variant of FedAvg that aggregates only the loss function part of the gradient or projects only when synchronizing requires $\Omega(1/\alpha^2)$ rounds of outer loop (i.e., communication), in order to converge to an $\alpha$ stationary point, i.e., increasing $E$ does *not* help, even if no noise is added.

This also says that DP-FedAvg is essentially the same as *stochastic* subgradient method in almost all locations of a piecewise linear objective function with gradient noise being $\mathcal{N}(0, \sigma^2/NI_d)$. The additional noise in DP-FedAvg imposes more challenges to the convergence. If we plan to run $T$ rounds and achieve $(\epsilon, \delta)$-DP, we need to choose $\sigma = \frac{\eta E G \sqrt{2T \log(1.25/\delta)}}{N\epsilon}$ (see, e.g., McMahan et al., 2018, Theorem 1). which results in a convergence rate upper bound of

$$\frac{GB(\sqrt{1 + \frac{2Td\log(1.25/\delta)}{N^2\epsilon^2}})}{\sqrt{T}} = O\left(\frac{GB}{\sqrt{T}} + \frac{\sqrt{d\log(1.25/\delta)}}{N\epsilon}\right),$$

for an optimal choice of the learning rate $E\eta$.

---

[1]One that outputs a solution in the subspace spanned by a sequence of subgradients.

The above bound is tight for stochastic subgradient methods, and in fact also information-theoretically optimal. The $GB/\sqrt{T}$ part of upper bound matches the information-theoretical lower bound for all methods that have access to $T$-calls of stochastic subgradient oracle (Agarwal et al., 2009, Theorem 1), while the second matches the information-theoretical lower bound for all $(\epsilon, \delta)$-differentially private methods on the agent level (Bassily et al., 2014, Theorem 5.3). That is, the first term indicates that there must be many rounds of communications, while the second term says that the dependence in ambient dimension $d$ is unavoidable for DP-FedAvg. Clearly, our method also has such a dependence *in the worst case*, but it is easier for our approach to adapt to the structure that exists in the data (i.e., high consensus among voting), as we will illustrate later. In contrast, it has larger impact on DP-FedAvg, since it needs to explicitly add noises with variance $\Omega(d)$.

Another observation is that when $N$ is small, no DP method with reasonable $\epsilon, \delta$ parameters is able to achieve high accuracy. This partially motivates us to consider the other regime that deals with instance-level DP.

### 3.3 Other Challenges

**Expensive Communication Cost:** Up-stream communication cost (Konečný et al., 2016), i.e., total transmitted updates from local agent to server, is another key concern in FL. For FedAvg, our convergence analysis suggests that increasing $E$ does not speed up the convergence. A high communication cost is expected till the model converges. CpSGD (Agarwal et al., 2018) is another DPFL method, aiming at reducing the communication cost by gradient quantization with binomial noise. However, sampling from binomial distribution can be difficult on devices, which prevents it from being practical in real-world scenarios.

**Network Complexity:** DP-FedAvg requires to clip gradient magnitude to $\tau$ at each coordinate in parameters, which is hard to scale up to large models, as the noise level increases proportional to the network capacity. To address this issue, recent works apply delicate clipping strategies (McMahan et al., 2018; Geyer et al., 2017) and reduce data dimension with PCA (Abadi et al., 2016). In this work, we propose to avoid such dimension dependence and empirically investigate how network architecture affects performance in various DPFL approaches.

## 4 Algorithm

We assume there are unlabeled data drawn from $\mathcal{D}_G$ at the server, which is public and accessible from any agent. The goal is to design an $(\epsilon, \delta)$-DP algorithm (either on the agent-level or instance-level) that outputs pseudo-labels for a subset of server's unlabeled data. Then a global model is trained in a semi-supervised way, using pseudo-labeled and unlabeled data.

**PATE-FL** In *PATE-FL* (Algorithm 1), each agent $i$ trains a local "teacher" model $f_i$ using its own private local data. For each "student" query $x_t$, every agent adds Gaussian Noise to her prediction (i.e., $C$-dim histogram), aggregates their noisy predictions via SMC and the label with the most votes is returned to the server as the "pseudo-label" of $x_t$. Similar to the original PATE, the idea behind the privacy guarantee is that by adding or removing any instance, it can *change* at most one agent's prediction. The same argument also naturally applies to *adding or removing one agent*. In fact we gain a factor of 2 in the stronger agent-level DP due to a smaller sensitivity (see the proof for details)! Another important difference is that in the original PATE, the teachers are trained on random splits of the data, while in our case, the agents are naturally present with different distributions. We propose to optionally use domain adaptation techniques to mitigate these differences when training the "teachers".

**Private-kNN-FL** Next we present how the teachers $f_i$ is constructed in *Private-kNN-FL* method (see Algorithm 2). Each agent has a data-independent feature extractor $\phi$. For every unlabeled query $x_t$, agent $i$ finds the $k_i$ nearest neighbor to $x_t$ from its local data by measuring their Euclidean distance in the feature space $\mathcal{R}^{d_\phi}$ and $f_i(x_t)$ outputs the frequency vector of the votes for these nearest neighbors. Subsequently, $f_i(x_t)$ from all agents are privately aggregated with the argmax of the noisy voting scores returned to the server.

Different from the original Private-kNN (Zhu et al., 2020), we apply kNN on each agent's local data instead of the entire private dataset. This distinction allows us to receive up to $kN$ neighbors while

---

**Algorithm 1** *PATE-FL*

**Input:** Noise $\sigma$, global data $\mathcal{D}_G$, $Q$ query

1: **for** $i$ in $N$ clients **do**
2: Train local model $f_i$ using $\mathcal{D}_i$
3: **for** $t = 0, 1, ..., Q$, pick $x_t \in \mathcal{D}_G$ **do**
4:   **for** each agent $i$ in $1, ..., N$ **do**
5:     $\tilde{f}_i(x_t) = f_i(x_t) + \mathcal{N}(0, \frac{\sigma^2}{N} I_C)$.
6:   **end for**
7:   $\tilde{y}_t = \arg\max_{y \in \{1,...,C\}} [\sum_{i=1}^N \tilde{f}_i(x_t)]_y$
8: **end for**
9: Train a global model $\theta$ using $(x_t, \tilde{y}_t)_{t=1}^Q$

---

**Algorithm 2** *Private-kNN-FL*

**Input:** Noise $\sigma$, global data $\mathcal{D}_G$, $Q$ query

1: **for** $t = 0, 1, ..., Q$, pick $x_t \in \mathcal{D}_G$ **do**
2:   **for** each agent $i$ in $1, ..., N$ **do**
3:     Apply $\phi$ on $\mathcal{D}_i$ and $x_t$
4:     $y_1, ..., y_k \leftarrow$ top-k closest labels
5:     $\tilde{f}_i(x_t) = \frac{1}{k}(\sum_{j=1}^k y_j) + \mathcal{N}(0, \frac{\sigma^2}{N} I_C)$
6:   **end for**
7:   $\tilde{y}_t = \arg\max_{y \in \{1,...,C\}} [\sum_{i=1}^N \tilde{f}_i(x_t)]_y$
8: **end for**
9: Train a global model $\theta$ using $(x_t, \tilde{y}_t)_{t=1}^Q$

---

bounding the contribution of individual agents by $k$. Comparing to PATE-FL, this approach enjoys a stronger instance-level DP guarantee since the sensitivity from adding or removing one instance is a factor of $k/2$ times smaller than that of the agent-level.

### 4.1 PRIVACY ANALYSIS

We provide our privacy analysis based on Renyi differential privacy (RDP) (Mironov, 2017). RDP inherits and generalizes the information-theoretical properties of DP, and has been used for privacy analysis in DP-FedAvg. We defer the background about RDP, its connection to DP and all proofs of our technical results to the appendix RDP section.

**Theorem 3** (Privacy guarantee). *Let PATE-FL and Private-kNN-FL answer $Q$ queries with noise scale $\sigma$. For agent-level protection, both algorithms guarantee $(\alpha, Q\alpha/(2\sigma^2))$-RDP for all $\alpha \geq 1$. For instance-level protection, PATE-FL and Private-kNN-FL obey $(\alpha, Q\alpha/\sigma^2)$ and $(\alpha, Q\alpha/(k\sigma^2))$-RDP respectively.*

This theorem says that both algorithms achieve agent-level and instance-level differential privacy. With the same noise injection to the agent's output, *Private-kNN-FL* enjoys a *stronger* instance-level DP (by a factor of $k/2$) compared to its agent-level guarantee, while PATE-FL's instance-level DP is *weaker* by a factor of 2.

**Improved accuracy and privacy with large margin:** Let $f_1, ..., f_N : \mathcal{X} \rightarrow \triangle^{C-1}$ where $\triangle^{C-1}$ denotes the probability simplex — the soft-label space. Note that both algorithms we propose can be viewed as voting of these local agents, which output a probability distribution in $\triangle^{C-1}$. First, let us define the margin parameter $\gamma(x)$ that measures the difference between the largest and second largest coordinate of $\frac{1}{N}\sum_{i=1}^N f_i(x)$.

**Lemma 4.** *Conditioning on the teachers, for each public data point $x$, the noise added to each coordinate of $\frac{1}{N}\sum_{i=1}^N f_i(x)$ is drawn from $\mathcal{N}(0, \sigma^2/N^2)$, then with probability $\geq 1 - C\exp\{-N^2\gamma(x)^2/8\sigma^2\}$, the privately released label matches the majority vote without noise.*

The proof (in Appendix) is a straightforward application of Gaussian tail bounds and a union bound over $C$ coordinates. This lemma implies that for all public data point $x$ such that $\gamma(x) \geq \frac{2\sqrt{2\log(C/\delta)}}{N}$, the output label matches noiseless majority votes with probability at least $1 - \delta$.

Next we show that for those data point $x$ such that $\gamma(x)$ is large, the privacy loss for releasing $\arg\max_j [\frac{1}{N}\sum_{i=1}^N f_i(x)]_j$ is exponentially smaller.

**Theorem 5.** *For each public data point $x$, the mechanism that releases $\arg\max_j [\frac{1}{N}\sum_{i=1}^N f_i(x) + \mathcal{N}(0, (\sigma^2/N^2)I_C)]_j$ obeys $(\alpha, \epsilon)$-data-dependent-RDP, where*

$$\epsilon \leq Ce^{-\frac{N^2\gamma(x)^2}{8\sigma^2}} + \frac{1}{\alpha - 1}\log\left(1 + e^{\frac{(2\alpha-1)\sigma^2}{2s^2} - \frac{N^2\gamma(x)^2}{16\sigma^2} + \log C}\right),$$

*where $s = 1$ for PATE-FL, and $s = 1/k$ for Private-KNN-FL.*

This bound implies that when the margin of the voting scores is large, the agents enjoy exponentially stronger (data-dependent) differential privacy guarantees in both agent-level and instance-level. In

| Datasets | # Agents | Methods | Accuracy | $\epsilon \downarrow$ |
|---|---|---|---|---|
| SVHN, MNIST $\rightarrow$ USPS | 200 | FedAvg | $87.6 \pm 0.1\%$ | - |
| | | FedAvg+DA | $86.9 \pm 0.1\%$ | - |
| | | DP-FedAvg | $76.3 \pm 0.3\%$ | 3.7 |
| | | DP-FedAvg+DA | $71.2 \pm 0.4\%$ | 3.6 |
| | | *PATE-FL* (Ours) | $83.8 \pm 0.2\%$ | 3.6 |
| | | *PATE-FL+DA* (Ours) | $92.5 \pm 0.2\%$ | 2.8 |
| CelebA | 300 | FedAvg | $84.9 \pm 0.1\%$ | - |
| | | DP-FedAvg | $83.2 \pm 0.1\%$ | 4.0 |
| | | *PATE-FL* (Ours) | $85.0 \pm 0.1\%$ | 4.0 |
| MNIST | 100 | FedAvg | $97.8 \pm 0.1\%$ | - |
| | | DP-FedAvg | $84.2 \pm 0.2\%$ | 4.3 |
| | | *PATE-FL* (Ours) | $95.1 \pm 0.3\%$ | 4.3 |

Table 1: **Agent-level DP Evaluation.** We compare the state-of-the-art DPFL methods with ours on the Digit and CelebA datasets. For $(\epsilon, \delta)$-DP setting, we set $\delta = 10^{-3}$ across all the methods.

other words, our proposed methods avoid the dependence on model dimension $d$ that are inherited in DP-FedAvg and can release models for free privacy cost when a high consensus among votes from local agents.

## 4.2 COMMUNICATION COST

Finally, regarding the communication issue, our proposed methods are *parallel* as each agent work independently without any synchronization. Overall, we reduce the up-stream communication cost from $d \cdot T$ floats (model size times $T$ rounds) to $C \cdot Q$ floats in one round.

## 5 EXPERIMENTAL RESULTS

We verify our *PATE-FL* for agent-level DP on Digit (LeCun et al., 1998; Netzer et al., 2011) and CelebA (Liu et al., 2015). Then, we evaluate *Private-kNN-FL* on Office-Caltech10 (Gong et al., 2012) and DomainNet (Peng et al., 2019a) for instance-level DP. Five independent rounds of experiments are conducted to report mean accuracy and its standard deviation. We use both labeled and unlabeled data on Digit datasets but only labeled data for all other datasets. We defer the experimental details to appendix.

### 5.1 EVALUATION ON AGENT-LEVEL DP

**Digit Datasets Evaluation**: MNIST, SVHN and USPS together as Digit datasets, is a controlled setting to mimic the real case, where distribution of agent-to-server or agent-to-agent can be different. We simulate 140 agents using SVHN with 3000 records each and 60 agents using MNIST with 1000 records each. USPS serves as unlabeled public data, where 3000 records can be accessed by the local agents and the remaining records are used for testing.

In Table 1, our methods *PATE-FL* and *PATE-FL+DA* are compared to private and non-private baselines. *PATE-FL+DA* is based on *PATE-FL*, where each agent model is trained with domain adaptation (DA) technique (Ganin et al., 2016). *FedAvg+DA* is the variant of *FedAvg* with the same DA technique. We observe:

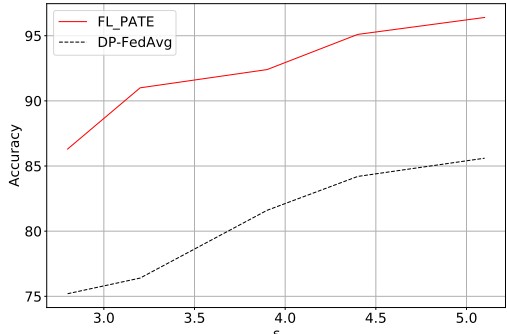

Figure 2: Privacy-accuracy tradeoff for MNIST dataset with Non-i.i.d partition. The $x$-axis is the privacy budget and the $y$-axis reports the corresponding accuracy.

(1) When the privacy cost $\epsilon$ of DP-FedAvg and *PATE-FL* is close, our method significantly improves the accuracy from $76.3\%$ to $83.8\%$. (2) The further improved accuracy $92.5\%$ of *PATE-FL+DA* demonstrates that our framework can orthogonally benefit from DA techniques, where it is highly uncertain yet for the gradient-based methods. (3) Both *FedAvg* and *DP-FedAvg* perform better than their DA variants. The possible reason might be that FL with domain adaptation is more closely

| Network | Methods | $A, C, D \rightarrow W$ (Acc.) | $\epsilon \downarrow$ | $A, C, W \rightarrow D$(Acc.) | $\epsilon \downarrow$ |
|---|---|---|---|---|---|
| | FedAvg | $90.5 \pm 0.1\%$ | - | $96.8 \pm 0.1\%$ | - |
| | DP-FedAvg | $28.1 \pm 0.7\%$ | 46.6 | $48.2 \pm 0.8\%$ | 47.1 |
| AlexNet | DP-FedSGD | $32.6 \pm 0.9\%$ | 4.1 | $48.3 \pm 0.9\%$ | 4.0 |
| | DP-FedSGD | $75.2 \pm 0.5\%$ | 12.4 | $83.7 \pm 0.6\%$ | 7.9 |
| | *Private-kNN-FL* (Ours) | $75.4 \pm 0.3\%$ | 3.9 | $84.3 \pm 0.3\%$ | 3.7 |
| | FedAvg | $96.5 \pm 0.1\%$ | - | $97.8 \pm 0.1\%$ | - |
| ResNet50 | DP-FedSGD | $25.8 \pm 0.6\%$ | 4.0 | $42.7 \pm 0.5\%$ | 3.9 |
| | *Private-kNN-FL* (Ours) | $86.3 \pm 0.4\%$ | 2.8 | $91.9 \pm 0.2\%$ | 2.0 |

Table 2: **Instance-level DP on Office-Caltech using different backbones.**

| | Clipart (Acc.) | $\epsilon \downarrow$ | Painting (Acc.) | $\epsilon \downarrow$ | Real (Acc.) | $\epsilon \downarrow$ |
|---|---|---|---|---|---|---|
| FedAvg | $81.8 \pm 0.2\%$ | - | $72.8 \pm 0.2\%$ | - | $82.0 \pm 0.3\%$ | - |
| DP-FedSGD | $44.2 \pm 0.2\%$ | 4.4 | $42.6 \pm 0.3\%$ | 4.6 | $39.1 \pm 0.6\%$ | 4.3 |
| DP-FedSGD | $55.6 \pm 0.2\%$ | 11.6 | $60.0 \pm 0.6\%$ | 14.6 | $55.1 \pm 0.6\%$ | 11.9 |
| *Private-kNN-FL* (Ours) | $55.8 \pm 0.6\%$ | 4.4 | $61.2 \pm 0.8\%$ | 4.7 | $55.5 \pm 0.7\%$ | 4.2 |

Table 3: **Instance-level DP on DomainNet.** We compare our method with DP-FedSGD and the non-private baseline FedAvg. Total number of local agents is 5. We set $\delta = 10^{-4}$.

related to multi-source domain adaptation (Peng et al., 2019b) than the traditional domain adaptation. In other words, averaging gradients of domain adaptation methods implies averaging different trajectories towards the server's distribution, which may not work in practice. How to improve DP-FedAvg variants with DA techniques remains an open problem.

**CelebA Dataset Evaluation:** CelebA is a 220k face attribute dataset with 40 attributes defined. 300 agents are designed with partitioned training data. We split 600 unlabeled data at server, and the rest 59,400 images are for testing. Detailed settings are referred to appendix. Consistent to Digits dataset, our method achieves clear performance gain by $1.8\%$ compared to DP-FedAvg while maintaining the same privacy cost.

**MNIST Dataset with Non-i.i.d Partition Evaluation:** In both CelebA and Digit experiments, we i.i.d partition each dataset into different agents. To investigate our proposed algorithm under a non-i.i.d partition scenario, we choose a similar experimental setup as (McMahan et al., 2017) did. We divide the training set of sorted MNIST into 100 agents, such that each agent will have samples from 6 digits only. This way, each agent gets 600 data points from 6 classes. We split 30% of the testing set in MNIST as the available unlabeled public data and the remaining testing set used for testing. As shown in Table 1, our method achieves consistent better performance than *DP-FedAvg*. Moreover, we plot the privacy-accuracy tradeoff in Figure 2. For every fixed privacy budget at the $x$-axis, we do a grid search on all hyperparameters (e.g., #queries and noise scale for *PATE-FL* and #communication round, noise scale for *DP-FedAvg*). In the figure, the accuracy of *PATE-FL* is consistently higher than *DP-FedAvg*.

## 5.2 EVALUATION ON INSTANCE-LEVEL DP

When agents are few, preserving privacy across agents becomes hard and meaningless. We then focus on preserving each instance's privacy, a.k.a instance-level DP. FedAvg is non-private baseline.

**Office-Caltech Evaluation:** Office-Caltech consists of data from four domains: Caltech (C), Amazon (A), Webcam(W) and DSLR (D). We pick one domain as server each time and the rest ones are for local agents (e.g., in $A, C, D \rightarrow W$, Webcam is treated as the server). We split 70% data from the server domain as public available unlabeled data while the remaining 30% data is used for testing. For *Private-kNN-FL*, we instantiate the public feature extractor using the network backbone without the classifier layer. Both AlexNet and Resnet50 are Imagenet pre-trained. We set $\sigma = 15$ for *Private-kNN-FL* with AlexNet and $\sigma = 25$ for ResNet50. To address the domain adaptation issue, each agent can choose $k$ to be smaller if they observe the domain gap is large, as a smaller $k$ implies a more selective set of neighbors. In our experiment, we set $k$ to be the 5% of the local data size (i.e., each agent returns the noisy top-5% neighbors' predictions).

We observe in Table 2, DP-FedSGD degrades when backbone changes from light load AlexNet to heavy load ResNet50, while ours is improved by $10\%$. It is because larger model capacity leads to more sensitive response to gradient clipping or noise injection. In contrast, our *Private-kNN-FL* avoids the gradient operation by label aggregation and can still benefit from the larger model

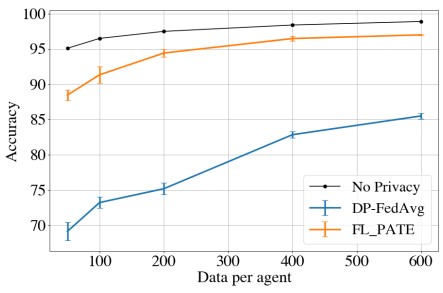 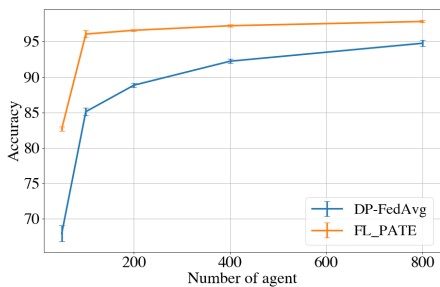

(a) Effect on the amount of data per agent     (b) Effect on the number of agent

Figure 3: Ablation study on amount of data per agent and number of agent. The non-private model, FedAvg, is served as the performance upper bound.

capacity. Again, our method achieves consistently better utility-privacy trade-off as maintaining same privacy cost and can achieve significantly better utility, or maintaining same utility and can achieve much low privacy cost.

**DomainNet Evaluation:** DomainNet contains 0.6 million images of 345 categories, ranging from six domains: Clipart, Painting, Real, Quickdraw, Infograph and Sketch. As a challenging dataset even for non-private setting (Peng et al., 2019c), we only consider seven fruit classes (apple, banana, grapes, strawberry, watermelon, pear, pineapple) for demonstration. Large domain shift exists between infograph/quickdraw and other domains (Peng et al., 2019c). Thus we only report results on cases where servers are chosen from Clipart, Painting and Real. Five domain data are assigned to five agents respectively. 70% of the left domain data is split for server and 30% rest for testing.

Table 3 compares our *Private-kNN-FL* method with DP-FedSGD. We observe that when the privacy cost $\epsilon$ is aligned close, our method outperforms DP-FedSGD by more than 10% in accuracy gain across all the three cases. When the accuracy is aligned close, our method saves more than 60% privacy cost, showing consistent advantage over DP-FedSGD.

### 5.3 ABLATION STUDY

In this section, we investigate the agent-level privacy-utility trade-off with respect to the number of agents and the volume of local data. MNIST is utilized for generality and simplicity. We randomly pick 1000 testing data as the unlabeled server data and the remaining 9000 data for testing. We adopt the model structure proposed in (Abadi et al., 2016) for both of our methods.

**Effect of *Data per Agent*:** We fix the number of agent to 100 and range the number of data per agent from $\{50, 100, 200, 400, 600\}$. By only relaxing the "data per agent" factor, we fairly tune the other privacy parameters for all the methods to its maximized performance. In Figure 3 (a), as "data per agent" increases, all the methods improves as the overall dataset volume increases. Our method achieves consistently higher accuracy over DP-FedAvg. The failure cases for both methods are when "data per agent" is below 50, which cannot ensure the well-trained local agent models. Label aggregation over such weak local models results in failure or sub-optimal performance.

**Effect of *Number of Agents*:** In Figure 3 (b), we vary $N \in \{50, 100, 200, 400, 800\}$ and set overall privacy budget fixed as $\epsilon = 5, \delta = 10^{-3}$. Following (Geyer et al., 2017), each agent has exactly 600 data, where data samples are duplicated when $N \in \{200, 400, 800\}$. We conduct grid search for each method to obtain optimal hyper-parameters. Our method shows clear performance advantage over DP-FedAvg. We also see DP-FedAvg gradually approaches our method as the number of agents increases.

## 6 CONCLUSIONS

In this work, we propose voting-based approaches for differentially private federated learning (DPFL) under two privacy regimes: agent-level and instance-level. We substantially investigate the real-world challenges of DPFL and demonstrate the advantages of our methods over gradient aggregation-based DPFL methods on utility, convergence, reliance on network capacity, and communication cost. Extensive empirical evaluation shows that our methods improve the privacy-utility trade-off in both privacy regimes.

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
