# OpenReview forum: "Voting-based Approaches For Differentially Private Federated Learning"
_ICLR.cc/2021/Conference — Reject_

### Official Review · AnonReviewer3 · 2020-10-21
**Review of Voting-based Approaches For Differentially Private Federated Learning**

**Rating:** 6
**Confidence:** 2

**Review:**

Objective of the paper:
The objective of the paper is to provide new federated learning algorithms based on aggregating labels from a voting scheme, instead of aggregating the gradients directly, to achieve more efficient differentially private algorithms.


Strong Points:
1)  It looks like the methods get strong results.
2)  Mix of theoretical results (in supplemental materials) and practical results.

Weak Points:
1)   The paper is not written as well as I would like.  Terms like clipping threshold are never actually defined in the main text.  Expressions like "each data".  Example 2 isn't well explained (why will it be clipped to 0)?  Statements like "lower bounded by O(BG/√T)"  (if it's a lower bound, shouldn't that be Omega not O?).  Tables are confusing (what is epsilon followed by a down-arrow?).
2)  The paper assumes a lot of background knowledge on the recent related work.  (I suppose that can't be helped, but it does make it a hard read.)  I don't understand how the "votes" work, in that I don't know what are the "labels" being voted on, or why the label with maximum vote is a reasonable approach.   (If I asked for a rating for a product and 10 people chose 1 but 9 people chose 8, 9, and 10, would I think the right rating is 1?)

Overall Rating:   I am left with the paper being borderline;  I will tentatively place it above the threshold based on the results looking good, and see what other reviewers suggest.

Questions for Authors:

Other Feedback:  I think in the end I was not the right reviewer for this paper, and I would defer to other reviewers (and raise my score as needed).  The paper suffers perhaps from being "too big" for the too-low page limit offered by ICLR.  The authors have tried to cram in a lot of experiments up front (and some theory in the supplemental materials), but the paper seems written essentially for people in the area.  Not enough up front explanation;  lots of assumed knowledge on the part of the reader in my mind.

---

> ### Author Response · Authors · 2020-11-20
> **Response to Reviewer 3**
>
> We thank the reviewer for the feedback. We acknowledge that details of some related work are omitted or deferred to the appendix, which we will try to include within space constraints.
>
> > **Q1.** “The clipping threshold is not defined in the main text.  Example 2 needs explanation. What is $\epsilon$ followed by a down-arrow?”
>
> - Clipping: in DP-FedAvg algorithm, we enforce clipping for per-agent update $\triangle_i$ by performing $\triangle_i / max(1, \|\|\triangle_i\|\|_2/ \tau)$. We clarified it in our paper now.
>
> - Example 2: we use a simplified example with two gradients, $\Delta_{1}$ larger than threshold and $\Delta_{2}$ less than threshold. Clipping will lead to $\Delta_{1}$ being biased and further averaging the two gradients leads to estimation bias. We have updated Example 2 in the paper.
>
> - Down-arrow:  it is a symbol to indicate that the lower value corresponds to the better performance for this $\epsilon$ evaluation.
>
>
> > **Q2.** “Statements like "lower bounded by O(BG/√T)." should be $\Omega$.
>
> Thanks for pointing this out. We have fixed it in the updated paper.
>
> > **Q3.** “ Why the label with maximum vote is a reasonable approach. (a rating for a product and 10 people chose 1 but 9 people chose 8, 9, and 10, would I think the right rating is 1?)”
>
> Majority voting of teachers is a key premise of PATE and private-kNN, which allows model-agnostic private learning and improves private-utility tradeoff. Intuitively, the voting represents the belief of each agent towards the true label of a candidate example, based on their own private data, which is a way to transfer knowledge from the distributed private data. Our results show that if the teachers are mostly unanimous, we incur a much smaller privacy loss.
>
> In the product rating example, our algorithm will output 1 with a larger probability but will have some probability outputting other labels as well. The randomness ensures differential privacy. The pseudo-label may not be the true label, even if the teachers agree.

---

### Official Review · AnonReviewer1 · 2020-10-28
**Simple and effective ideas, some details are missing**

**Rating:** 5
**Confidence:** 4

**Review:**

Summary:
The paper proposes two approaches (i.e., PATE-FL and Private-kNN-FL) to train a differentially private global model in a federated setting based on [1] and [2]. In PATE-FL, each client first trains a teacher model using their local dataset. The teacher models are used to make noisy predictions on a public dataset. Then, the public dataset with predicted labels is used to train a final model. In Private-kNN-FL, instead of training a teacher model, the prediction depends on the labels of the nearest neighbors. The experiments show that PATE-FL and Private-kNN-FL can outperform DP-FedAvg in terms of agent-level DP and instance-level DP, respectively.

[1] Scalable Private Learning with Pate   ICLR 2018

[2] Private-kNN: Practical Differential Privacy for Computer Vision   CVPR 2020

Pros:

(1) The studied problem is interesting and important.

(2) The proposed approaches have quite good performance compared with DP-FedAvg.

Cons:

(1) The work does not have sufficient algorithmic contributions. PATE-FL simply applies PATE [1] in the federated setting without the data splitting since the data are naturally partitioned. Also, Private-kNN-FL uses Private-kNN [2] in local clients to label the public dataset. The algorithms are an extension of existing algorithms in the federated setting without proposing new techniques. The theoretical analysis seems can be easily derived from the existing theorems with a modified sensitivity.

(2) The proposed approach lacks some important details. In Private-kNN-FL, a data-independent feature extractor is used. However, the authors did not mention how to get the extractor. In federated learning scenarios, the training data are usually sensitive and thus it is challenging to get a good and public feature extractor for the data. If the quality of the feature extractor is bad, then the labeling may be quite bad especially when the data are non-i.i.d. distributed (which is a common case in federated learning). Also, it is not clear how to set $k$ for Private-kNN-FL.

(3) The experiments are not clear and solid enough. Some experimental details are missing. Also, more experiments are needed. Please see the detailed comments.

Detailed comments:

(1) The experimental details are missing: the number of communication rounds for FedAvg, the value of $k$, the feature extractor, the test dataset of Digit task.

(2) I’d like to see the experiments given different numbers of queries/$\varepsilon$. The authors may add a figure to show the accuracy versus $\varepsilon$ of all approaches.

(3) The experiments for agent-level DP evaluation can be further improved. It seems that each dataset is i.i.d. partitioned into different clients. The authors can try non-i.i.d. partition [3,4], which is a key feature in federated learning.

[3] Communication-Efficient Learning of Deep Networks from Decentralized Data 		AISTATS 2017

[4] Federated Learning with Matched Averaging  	ICLR 2020


=======Post-rebuttal comments=========

Thanks for the authors' response. However, my main concerns on the feature extractor and the contributions are still not well addressed. Moreover,

(1) The authors agree that a good feature extractor is usually hard to obtain in practice. The feature extractor will significantly limit the applicability of the proposed approach.

(2) I found that the authors use ImageNet pretrained models in the experiments. This setting is weird. In my knowledge, the pretrained models are seldom used in the existing federated learning studies. The model may already be good enough before training and thus cannot well show the effectiveness of the algorithms.

---

> ### Author Response · Authors · 2020-11-20
> **Response to Reviewer 1**
>
> We thank the reviewer for the feedback. We have updated the paper with more experimental details and added two new experiments as suggested.
>
>
> > **Q1.** “work algorithmic contributions.”
>
> Please refer to the clarification at the beginning of the rebuttal and the remark on page 1 of the updated paper.
>
>
> > **Q2.** “The theoretical analysis seems can be easily derived from the existing theorems with a modified sensitivity.”
>
> While Theorem 3 comes from the Gaussian Mechanism with different sensitivity, our analysis of sensitivity itself is new. In fact, we gain by reducing privacy cost from original instance-level DP (PATE) by 50% in the stronger agent-level DP. Moreover, Theorem 5 is novel. To show the privacy loss is exponentially smaller when there is a high margin, our proof involves two steps:
>   1) Show the output labels match noiseless majority votes with probability exponentially close to 1 (Lemma 12 in appendix);
>   2) Bound the data-dependent RDP condition on the high probability event (Lemma 13 in appendix). More theoretical analysis can be found in appendix E.2.
>
>
> > **Q3.** “The labeling may be biased as the feature extractor can be bad when data are non-i.i.d. How to set k for private-kNN-FL?”
>
> We agree that obtaining a good feature extractor could be difficult in practice. We simply use a pretrained ImageNet model, but the iterative feature updating and labeling mechanism of Private-kNN then achieves substantially improved performance within just 3-5 iterations.
>
> Our Private-kNN-FL allows more flexible choices for k, where DP-FedAvg requires the same network architecture for each local agent. In experiments, each agent returns the top 5% of all its neighbors’ predictions.
>
>
> > **Q4.** “Some experimental details are missing, e.g., how to get the extractor, #communication round for FedAvg and the test dataset in digit task.”
>
> We defer some details (e.g, the feature extractor, test dataset) to the appendix due to space limitations. We have updated the paper to incorporate more experimental details.
>
> We instantiate the data independent feature extractor using the network backbone without the classifier layer. For example, both AlexNet and Resnet50 in Table 2 are Imagenet pre-trained. When evaluating the instance-level DP using ``AlexNet” backbone, we construct the feature extractor by dropping the classifier layer in Alexnet.
>
> For the digits task test set, we split 3000 records from USPS as the unlabeled public data and the remaining records (~4000 records) are used for testing.
>
> For communication rounds in FedAvg, we run FedAvg until the algorithm converges.  We report the pre-defined maximum number of communication rounds.
>
> | | Digit | CelebA | Office-Caltech | Domain-Net |
> |:-:|:-:|:-:|:-:|:-:|
> | # Communication round | 150   | 100    | 50 | 200  |
>
> Nots that the proposed methods only require one round communication. More discussions on parameter setting for DP-FedAvg / DP-FedSGD can be found in Section F of the appendix.
>
> > **Q5.** “The authors can try non-i.i.d. partition for agent-level DP, which is a key feature in federated learning.”
>
> Thanks for the suggestion. We have updated the paper with a non-i.i.d partition experiment (the third row in Table 1).
>
> We choose a similar experimental setting as in the AISTATS-17 paper (Communication-Efficient Learning of Deep Networks from Decentralized Data ) as the reviewer suggests. We divide the training set of sorted MNIST into 100 agents, such that each agent has samples from 6 digits only under the non-i.i.d. assumption. Each agent gets 600 data points from 6 classes. We split 30% of the testing set in MNIST as the available unlabeled public data and the remaining testing set used for testing. As in Table 1, our method still achieves consistently better performance than DP-FedAvg, i.e., PATE-FL (ours) achieves acc. 95.1% +- 0.3% while DP-FedAvg achieves acc. 84.2%+-0.2% with the same privacy cost 3.7.
>
>
> > **Q6.** “Authors may add a figure to show the accuracy versus $\epsilon$ of all approaches.”
>
> Thanks for the suggestion. We have updated the paper with a new privacy-accuracy tradeoff figure. The experimental setting is the same as the non-i.i.d partition experiment with MNIST dataset. For each fixed privacy budget at the x-axis, we do a grid search on all hyperparameters (e.g., #queries and noise scale for PATE-FL  and  #communication round, noise scale for DP-FedAvg). In Figure 2, the accuracy of PATE-FL is consistently higher than DP-FedAvg, which implies that our method is advantageous in a wide range of $\epsilon$.

---

### Official Review · AnonReviewer4 · 2020-11-04
**A new private federated learning algorithm**

**Rating:** 4
**Confidence:** 4

**Review:**

Federated learning enables distributed clients to train a model without sharing the data with each other. This is typically achieved by a gradient descent type algorithm such as federated averaging. The paper argues that federated learning via gradient updates has issues and proposes to use a voting based method for training machine learning models using unlabeled global data.

I have several concerns about the paper:

1. The novelty in the new algorithms is not clear.  PATE-FL and Private-kNN-FL are largely similar to the original PATE and Private-kNN algorithms.
2. The paper argues that the existing gradient descent type algorithms suffer from slow convergence. However, recent works such as SCAFFOLD (https://arxiv.org/pdf/1910.06378.pdf) provide fast convergence rates for gradient based federated learning. It would be good to add comparisons with these new algorithms.
3. In Example 2, the problem can be mitigated by setting the threshold to \tau + \alpha. Hence its not clear if this is an inherent issue in gradient based methods or if it is just an issue with hyperparameter tuning.
4. Unlike standard works, the proposed algorithms assume the existence of a global unlabeled dataset, which is not always feasible. Furthermore, the approach inherently requires that the global data is similar to the private datasets, which has not been quantified. For a fair comparison in experiments, one should consider variations of federated averaging with the global public data.

---

> ### Author Response · Authors · 2020-11-20
> **Response to Reviewer 4**
>
> Thank you for the comments. We have updated the paper with new experiments on DP-FedAvg with the global public data.
>
> > **Q1.** “Novelty of the new algorithms.”
>
> Please refer to the clarification at the beginning of the rebuttal and the remark on page 1 of the updated paper.
>
> > **Q2.** “Add comparisons with SCAFFOLD.”
>
> SCAFFOLD does not handle differential privacy. It’s interesting future work to consider how SCAFFOLD may be extended to a DP setting, which we outline in an updated discussion in Sec 3.2.
>
> We have looked closely into their technical results and we are happy to report that **there are no contradictions**.  At the high level,  the SCAFFOLD paper shows that with a smoothness assumption, using multiple local steps reduces the stochastic gradient’s variance without introducing substantial bias, while we show that in a more general family of problems there are hard cases where multiple local steps (even if the variance is 0 in all of them) are essentially increasing learning rates in the outer loop, hence satisfying the same convergence lower bound for all Krylov space methods.
>
> Acknowledging SCAFFOLD with better per-round convergence performance for FL methods, we would like to emphasize:
>   - We focus on convergence of DPFL methods, not FL methods
>   - Counting iteration rounds, DP-FedAvg requires many rounds while Private-kNN-FL and PATE-FL require only one round.
>
> > **Q3.** “In Example 2, is it an inherent issue or hyperparameter tuning?”
>
> It is an issue of clipping. Clipping will result in gradient estimate bias, i.e., when sample with gradient larger than threshold is clipped, further averaging with some other gradients will result in a loss of gradient signal. We have updated Example 2 to make it clearer. Moreover, a larger clipping threshold requires a larger scale of noise injection, which also suggests clipping is an inherent issue in gradient-based methods. The choice of threshold will not be data-dependent unless the hyper-parameter tuning is done in a differentially private fashion.
>
>
> > **Q4.** “global unlabeled dataset may not always be feasible. The proposed approach requires that the global data is similar to the private dataset.”
>
> Indeed, public datasets may not always be available. However, we study the new problem of federated learning with a DP constraint and follow the standard DP settings of PATE and Private-kNN where public data is leveraged. Agnostic FL studies such as [1, 2] also assume unlabeled server data.
> We emphasize again, as stated in the paper and shown experimentally in Tables 2 and 3, that distribution of global data at server can be **different** from that of private data at local agents.
>
> > **Q5.** “One should consider variations of federated averaging with the global public data for a fair comparison.”
>
> **We aim to study how DP can enhance general FL frameworks**. To study different FL variants for our DP framework is outside the scope of this paper, but possible future work. We self-implement the baseline DP-FedAvg on top of FedAvg. Our purpose is to compare the voting based methods (ours) to the gradient based methods (DP-FedAvg).
>
> [1] Agnostic federated learning.  Mehryar Mohri, Gary Sivek, and Ananda Theertha Suresh.
>
> [2] Federated adversarial domain adaptation. Xingchao Peng, Zijun Huang, Yizhe Zhu, and Kate Saenko.
>
> [3] Domain-adversarial training of neural networks. Yaroslav Ganin, Evgeniya Ustinova, Hana Ajakan, Pascal Germain, Hugo Larochelle, Francois Laviolette, Mario Marchand, and Victor Lempitsky.

---

### Official Review · AnonReviewer5 · 2020-11-06
**Marginally above acceptance**

**Rating:** 6
**Confidence:** 2

**Review:**

Voting-based approaches for DP

1/ Summary of the paper

This paper addresses the problem of learning a private model in an FL setting. It assumes that the server has an unlabelled dataset on which learning has to be performed. The paper proposes two extensions of known algorithms, PATE and Private-KNN in this setting. In both cases, up to variations, the approach is "one-shot" and amounts to getting consensus labels for the public dataset from the different FL agent, the consensus being obtained with DP guarantees. Then, a model is trained on the server using these consensus labels.
The paper provides Rényi DP guarantees for both algorithms. Experiments on real datasets in 2 settings (many agents and few ones) show an improved performance over baseline DP FL algorithms.

2/ Strong and weak points

- The paper makes a very good distinction of the number of agents and the DP guarantees it implies in both settings, with the notion of instance-level (i.e. at the individual sample level) vs agent-level (i.e. client) DP.
- The paper is well-motivated, and section 3 is very helpful in this regard (even if I have some remarks, see additional comments)
- The paper is well written and easy to follow

- A weakness of the paper is its limited algorithmic novelty: as far as I understand, PATE already considered N subsets of a large dataset, while here the subsets are already provided by the agents; Private-KNN-FL is more novel insofar as private-KNN only considered a single dataset, but here as well the jump from 1 to N is the only novelty.
- Another weakness of the paper is its experimental validation, see supporting arguments.

3/ Recommendation

Given the limited algorithmic novelty and the limitations of the experimental validation, I think this paper is marginally above acceptance threshold despite its interesting viewpoint and theoretical contributions.

4/ Supporting arguments

- All results in Table 1 and Table 2 rely on domain adaptation approach, with the final model being tested on a domain held by the server. While the proposed approaches really train a model on this domain (at least for the samples, not the labels), the baselines FedAvg, DP-FedAvg and DP-FedSGD are not designed for this setting by default, as they minimise the average losses of the clients and never see the server distribution. Out-of-domain generalisation is an issue of its own in FL, and the mix of both issues at the same time makes it impossible to really compare results with one another, rendering all these results less significant.
- Theorem 3 provides privacy guarantees of both approaches in the agent-level and instance-level DP cases, allowing to compare the proposed approaches with one another. However, in the experimental section, PATE-FL is compared to a baseline agent-level method, and Private-KNN-FL approach is compared with instance-level methods. Private-KNN-FL and PATE-FL are never compared with one another experimentally, which would have been helpful to understand when one is better than the other (especially for the ablation experiments of Fig 2).

5/ Questions and comments

- Could you detail the experimental methodology to get results from table 3? In particular, from which domain were the different agents extracted from? Does each column correspond to a different training?
- What Domain adaptation method is used for SVHn and MNIST in Table 1? What is the corresponding performance of FedAvg and DP-FedAvg when applying it to them as well?
- In Section 3.1, example 2 appears a bit artificial to me insofar as it is more the heterogeneity of the different clients (with positive and negative x_i) which yields a null gradient than the clipping. In particular, this is a low-dimensional example, whereas in high-dimensions it is well known that the magnitude of the gradients play less of a role, see e.g. the effectiveness of SignSGD, Adam or TernaryGrad compression methods in the FL setting.

---

> ### Author Response · Authors · 2020-11-20
> **Response to Reviewer 5**
>
> Thank you for the helpful feedback. We have updated the paper with experiments on DP-FedAvg with global public data.
>
> > **Q1.** Algorithmic novelty.
>
> Please refer to the clarification at the beginning of the rebuttal and the remark on page 1 of the updated paper.
>
> > **Q2.** “PATE-FL for agent-level and Private-kNN-FL for instance-level experiments. They two are not compared with one another.”
>
> Our focus is on comparing voting-based and gradient-based methods, so we wish to avoid confusion by suggesting comparisons among PATE-FL and Private-kNN-FL. Having said that, we make the above choice as:
>   - Private-kNN-FL enjoys a stronger instance-level DP (by a factor of k) compared to PATE-FL (Theorem 3). It is straightforward to apply Private-kNN-FL for instance-level DP experiments.
>   - PATE-FL for agent-level experiments made for a clearer exposition.
>
> > **Q3.** “In Table 3, from which domain were the different agents extracted?”
>
> DomainNet consists of data from six domains. For each column in Table 3 (e.g., the Clipart column), we consider Clipart domain as the server while data from the remaining five domains are assigned to five local agents.
>
> > **Q4.** “All results in Table 1 and Table 2 rely on domain adaptation approach, … the baselines FedAvg, DP-FedAvg and DP-FedSGD are not designed for this setting.”
>
> To distinguish our method with the baselines, we tried the following two ways.
>
>   - Table 1, CelebA and MNIST do not use domain adaptation, the same setting as the baselines. “SVHN,MNIST->USPS” uses domain adaptation. Our method yields consistent advantages, even comparable to non-DP FedAvg.
>   - We tried to apply domain adaptation to improve DP-FedAvg, so that the setting is the same as our method. (To our best, there is no existing work on DP-FedAvg to handle domain adaptation.)
>
> Before submission, we implemented DP-FedAvg with Domain Adversarial Neural Network (DANN) [1]. Evaluation is conducted following Table 1, but leads to worse performance than DP-FedAvg (e.g., with same $\epsilon$, DP-FedAvg  is 76.3%, DP-FedAvg+DA is 71.2% on SVHN,MNIST->USPS). Thus we did not report that in our original submission (updated in Table 1 with discussions at the bottom of page 7).
>
> [1]  Domain-adversarial training of neural networks. Yaroslav Ganin, Evgeniya Ustinova, Hana Ajakan, Pascal Germain, Hugo Larochelle, Francois Laviolette, Mario Marchand, and Victor Lempitsky.
>
> > **Q5.** Domain adaptation method used for SVHN and MNIST?  and the performance of FedAvg and DP-FedAvg.
>
> DANN is used for domain adaptation. Performance is updated in Table 1, i.e.
> FedAvg achieves 87.6% acc., FedAvg + DA is with 86.9% acc..  Under the same privacy budget, DP-FedAvg achieves acc. 76.3% while DP-FedAvg + Da achieves 71.2%.
>
> > **Q6.** “Example 2 appears a bit artificial”
>
> We have updated Example 2 in paper to show that: Under the most simplified condition, we consider two gradients, $\Delta_{1}$ larger than threshold and $\Delta_{2}$ less than threshold. Clipping will lead to $\Delta_{1}$ being biased and further averaging the two gradients leads to estimation bias. This simple example can be extended to more general cases, leading to the same conclusion.

---

### Author Response · Authors · 2020-11-20
**Clarification on novelty and contributions**

We thank all the reviewers for their feedback. We have updated the paper to incorporate the suggestions, with updates in red.

- Major Additions:
  - Experiment of FedAvg / DP-FedAvg with domain adaptation (Table 1). (R4 and R5)
  - Experiment of agent-level DP with non-i.i.d partitioned data and a privacy-accuracy tradeoff figure (Table 1 and Figure 2). (R1)
  - A remark on novelty in the introduction.

- Important Minor Changes:
  - We have updated “Example 2” in the paper to clarify that __clipping__ is an inherent issue in gradient based methods.

### Remark of Algorithmic Novelty

While our proposed PATE-FL and Private-kNN-FL share algorithmic traits with PATE and Private-kNN, we extend them in non-trivial ways:

1) Federated Learning with Differential Privacy (DPFL) introduces new constraints and challenges:
  - Key algorithmic components (privacy amplification by sampling,  noisy screening) in PATE or Private-kNN do not hold in DPFL.
  - An attacker in DPFL can eavesdrop on all network traffic, while only seeing the final models in standard private learning.

2) **Agent-level DP** is first proposed in PATE-FL and Private-kNN-FL as a novel notion, where PATE and Private-kNN only provide instance-level DP.

3) PATE-FL and Private-kNN-FL can address **non-I.I.D** data distributions from local agents, whereas PATE assumes I.I.D distribution for each teacher model.

The table below compares our DPFL methods to prior DP works, showing that we are the first to study the key technical challenges (privacy and communication) that arise in DPFL.

|  | PATE | PATE-FL | Private-kNN | Private-kNN-FL |
|:-|:-:|:-:|:-:|:-:|
|Screening | Yes | N/A | Yes | N/A |
| Subsampling | N/A | N/A| yes | N/A|
| Data I.I.D | Yes | No | Yes | No |
| DP Notion (Add/Remove what?) | Instance | Instance / Agent | Instance | Instance / Agent |
| Attacker observes | Pseudo-labels | All network traffic | Pseudo-labels | All network traffic |

---

### Decision · Program_Chairs · 2021-01-07
**Final Decision**

**Decision:**

Reject

**Comment:**

This paper adapts the semi-supervised DP learning methods based on voting to FL. Specifically, PATE and private-kNN. The adaptation is fairly straightforward as those methods rely on averaging of votes a primitive that is a standard part of FL. The framework assumes that unlabeled data from the same distribution is available to the server, a very strong assumption.
As pointed out in the reviews, the empirical evaluation has a number of major issues. For example, the comparison is with fully-supervised SGD based techniques instead of a gradient-based semi-supervised approach.